# Pleiotrophin and the Expression of Its Receptors during Development of the Human Cerebellar Cortex

**DOI:** 10.3390/cells12131733

**Published:** 2023-06-27

**Authors:** Margarita Belem Santana-Bejarano, Paula Romina Grosso-Martínez, Ana Graciela Puebla-Mora, María Guadalupe Martínez-Silva, Mario Nava-Villalba, Ana Laura Márquez-Aguirre, Daniel Ortuño-Sahagún, Marisol Godínez-Rubí

**Affiliations:** 1Laboratorio de Patología Diagnóstica e Inmunohistoquímica, Centro de Investigación y Diagnóstico en Patología, Departamento de Microbiología y Patología, CUCS, Universidad de Guadalajara, Guadalajara 44340, Mexico; santana.bejarano@gmail.com (M.B.S.-B.); rominagrossmar@gmail.com (P.R.G.-M.); graciela.puebla@academicos.udg.mx (A.G.P.-M.); 2Doctorado en Ciencias en Biología Molecular en Medicina, Centro Universitario de Ciencias de la Salud, Universidad de Guadalajara, Guadalajara 44340, Mexico; 3Departamento de Anatomía Patológica, Centro Médico Nacional de Occidente, Instituto Mexicano del Seguro Social (IMSS), Guadalajara 44340, Mexico; lupita.doctora694@gmail.com; 4Centro de Investigación y Diagnóstico en Patología, Departamento de Microbiología y Patología, CUCS, Universidad de Guadalajara, Guadalajara 44340, Mexico; mario.nava@academicos.udg.mx; 5Unidad de Biotecnología Médica y Farmacéutica, Centro de Investigación y Asistencia en Tecnología y Diseño del Estado de Jalisco A.C. (CIATEJ), Guadalajara 44270, Mexico; amarquez@ciatej.mx; 6Laboratorio de Neuroinmunobiología Molecular, Instituto de Investigación en Ciencias Biomédicas (IICB), CUCS, Universidad de Guadalajara, Guadalajara 44340, Mexico; 7Departamento de Morfología, CUCS, Universidad de Guadalajara, Guadalajara 44340, Mexico

**Keywords:** human, cerebellum, development, pleiotrophin, RPTPZ1, ALK, NRP-1

## Abstract

During embryonic and fetal development, the cerebellum undergoes several histological changes that require a specific microenvironment. Pleiotrophin (PTN) has been related to cerebral and cerebellar cortex ontogenesis in different species. PTN signaling includes PTPRZ1, ALK, and NRP-1 receptors, which are implicated in cell differentiation, migration, and proliferation. However, its involvement in human cerebellar development has not been described so far. Therefore, we investigated whether PTN and its receptors were expressed in the human cerebellar cortex during fetal and early neonatal development. The expression profile of PTN and its receptors was analyzed using an immunohistochemical method. PTN, PTPRZ1, and NRP-1 were expressed from week 17 to the postnatal stage, with variable expression among granule cell precursors, glial cells, and Purkinje cells. ALK was only expressed during week 31. These results suggest that, in the fetal and neonatal human cerebellum, PTN is involved in cell communication through granule cell precursors, Bergmann glia, and Purkinje cells via PTPRZ1, NRP-1, and ALK signaling. This communication could be involved in cell proliferation and cellular migration. Overall, the present study represents the first characterization of PTN, PTPRZ1, ALK, and NRP-1 expression in human tissues, suggesting their involvement in cerebellar cortex development.

## 1. Introduction

In humans, cerebellar development starts during the fourth gestational week (gw), and it continuously changes until it matures [1]. This process includes cell proliferation, migration, differentiation, and apoptosis events. Specifically, the cerebellum arises from the rhombencephalon, and it undergoes a complex maturation process that finishes at 2 years old [2,3]. In the latest human fetal stage, the cerebellar cortex is structured by the external granule cell layer (EGL), molecular layer (ML), Purkinje cell layer (PCL), and internal granule cell layer (IGL). When a human is approximately two years old, granule cells culminate their migration from the EGL to the IGL via Bergmann glia [4,5,6]. The Bergmann glia have an astroglial phenotype, which expresses glial fibrillary acidic protein (GFAP) during human fetal development [7,8,9,10]. Any disruption in these processes can lead to cerebellar pathology in the form of medulloblastoma, the molecular subgroup and prognosis of which depend on the specific developmental abnormalities [11]. Additionally, it has been demonstrated that extracellular growth factors, such as pleiotrophin (PTN) and midkine (MK), play an important role in rodent cerebral and cerebellar cortex ontogenesis, with special emphasis on the postnatal stage [12,13,14,15,16,17]. Nevertheless, its expression has not been demonstrated in the developing human cerebellum. On the other hand, alterations in PTN are related to tumor cell proliferation and prognosis of neoplasms, such as glioblastoma, testicular germ cell neoplasms, and hepatocellular carcinoma [18,19,20,21]. 

PTN is a protein composed of 168 amino acids (a.a.) that is secreted into the extracellular matrix [22,23], where it has a signaling function for neural survival, differentiation, and migration molecules during mouse and rat embryonic induction [12,24], and is beginning to be recognized as a neuromodulatory cytokine [17]. This signaling function is mediated by different receptors, such as receptor-type protein-tyrosine kinase Z receptor (PTPRZ1), anaplastic lymphoma kinase receptor (ALK), and neuropilin-1 receptor (NRP-1) [25,26,27,28]. PTPRZ1 is a chondroitin sulfate proteoglycan that has intracellular phosphatase activity. It has been found in Purkinje cells (PCs) and Bergmann glia in postnatal rat cerebellar cortices and neural axon fibers during growth [16,29]. Additionally, the PTN–PTPRZ complex induces PTPRZ1 tyrosine phosphatase inactivation in U373-MG cells, and this signaling pathway is involved in neural cell migration in the brain cortex [30,31]. On the other hand, ALK is a receptor tyrosine kinase that is structurally similar to the insulin receptor subfamily [32]. The mRNA and protein expression of ALK has been described in the neural cells of the diencephalon, neuroepithelium, thalamus, midbrain, and spinal cord in mouse development [33]. Moreover, the interaction between PTN and ALK depends on PTPRZ1 inhibition by the PTN–PTPRZ1 complex [34]. NRP-1 is a 130 KDa non-tyrosine kinase receptor, and it has a transmembrane single-pass structure [35]. NRP-1 has been described as a chemo-repulsive molecule involved in the neuronal growth cone in the peripheral nerves [36,37]. Furthermore, NRP-1 was found expressed in vitro in Schwann cells and hippocampal neurons [37]. 

Even though these molecules are reported in nervous system development, most knowledge is based on animal models, and their cerebellar cortex localization is still debated. For this reason, this study aims to describe the expression of PTN and its receptors in the human fetal and neonatal cerebellar cortex.

## 2. Materials and Methods

### 2.1. Human Paraffin-Embedded Tissues

The present study was approved by the Ethics Committee of Centro Médico Nacional de Occidente (México) and Centro Universitario de Ciencias de la Salud, Universidad de Guadalajara, Jalisco, México (approval numbers: R-2021-1301-123 and CI-02021, respectively). Five fetal samples and one neonatal sample were collected from pathology autopsy studies between 2013 and 2015. For human fetal samples, there were three females at 31, 37.4, and 38 gw; two males at 17 and 36.6 gw. For human neonatal samples, there was one female at 38 gw. Any infectious process was discarded in the clinical history. 

### 2.2. Tissue Preparation and Histological Staining 

Fetal ages were calculated from the first day of the mother’s last menstrual period and fetal somatometry study. The human fetal cerebellum was fixed in 4% paraformaldehyde with a pH of 7.4. They were dehydrated with ethanol and paraffin embedded. Then, 3 µm thick sections were made, and hematoxylin and eosin staining (H&E) was performed for histological examination.

### 2.3. Immunohistochemistry (IHC) 

The analyses were performed on paraffin-embedded tissues from the fetal cerebellar cortex. The paraffin block was cut into 3 µm thick sections, deparaffinized, and rehydrated. For all IHC analyses, a Novolink^TM^ Polymer Detection System was used (RE7290-K, Leica BIOSYSTEMS). Antigen retrieval was performed using citrate buffer solution (pH, 6.0) to analyze PTN, PTPRZ1, and ALK; EDTA buffer solution (pH, 8.0) was used for NRP-1 antigen retrieval. Endogenous peroxidase was blocked with 3% (*v*/*v*) H_2_O_2_. Novocastra^TM^ protein block was used for 30 min to reduce the non-specific binding of primary antibodies and polymers. Anti-PTN (1:100, sc-74443, Santa Cruz) and anti-PTPRZ1 (1:100, sc-33664, Santa Cruz) primary antibodies were incubated for 60 min; anti-ALK (1:10, ACI 3041 A, Biocare) was hatched for 90 min; anti-NRP-1 (1:250, Ab81321, Abcam) and anti-GFAP (1:250, 258R, Cell Marque) were incubated for 30 min. The tissues were then treated with post-primary (rabbit anti-mouse IgG) and Novolink ^TM^ polymer solutions (anti-rabbit poly-HRP-IgG). The chromogen was 5% 3,3′-diaminobenzidine in a buffer solution containing 0.1% H_2_O_2_. The sections were counterstained with hematoxylin, dehydrated, and cover slipped. The specificity of antibody labeling was previously verified in several control tissues (Appendix A). Seminoma tissue was utilized as a PTN-positive control tissue; cecal appendix and cerebral cortex as PTPRZ1-positive control tissues; cecal appendix as an NRP-1-positive control tissue; cerebellar cortex as an ALK-positive control tissue. The same tissues without primary antibodies were used as the negative control. Immunohistochemical staining was evaluated by one pathologist. 

### 2.4. Histological Analysis

The tissue was digitalized using Aperio LV1 (Leica Biosystem Imaging Inc., Deer Park, IL, USA). The analysis was conducted on a total of 4 different locations in each sample using Qupath (open-source software for digital pathology image analysis) [38]. EGL cell density was counted at 5000 µm^2^ per section using hematoxylin detection as a measure of total cells. The thickness of the EGL and ML was defined with the spatial analysis function. Purkinje cell area was automatically determined by the software using an ellipse selection form. The staining intensity measurement in immunohistochemistry was conducted with a set cell intensity classification function according to DAB subcellular localization. All staining intensity measurements were evaluated as a percentage of four different fields analyzed and reported using three categories: 1+ (weak), 2+ (moderate), and 3+ (strong) (Appendix A).

## 3. Results

The cerebellar cortex was analyzed for all human samples. The majority of them were female (67%), with a median age of 37 gw (range: 17–38 gw). The cause of death of the unique neonatal sample was respiratory failure. The clinical characteristics of the human cases are summarized in Table 1. 

### 3.1. The Topological Architecture of the Developing Human Cerebellar Cortex

Cerebellar cortex cellularity was observed by using hematoxylin and eosin staining. The EGL was constantly detected in all gestational ages analyzed. The cell thickness and density showed variations at different gestational ages. The age with the highest EGL cell thickness and density of EGL was 36.6 gw, and the lowest density was at 17 gw (Table 2, Figure 1A,B,F). Additionally, at 31 gw, the PC bodies could be easily detected in comparison to those at 17 gw. PCs increase the surface area of their soma as the gestational age increases, and this tendency is maintained in the postnatal stage, wherein the largest cell size for this type of cell is recorded. The molecular layer was not visible at 17 gw and only became evident at 31 gw; its maximum thickness could be seen at 36.6 gw, with a drastic reduction in cases of higher gestational age (Figure 1A–F, Table 2). From 36.6 gw, the IGL was noticeable (Figure 1C). 

Interestingly, the fetal and neonatal samples at 38 gw showed histological differences. The EGL density was higher in the fetal cerebellar cortex than in the neonatal cerebellar cortex. The Purkinje axonal projections were more prominent in the neonatal cerebellar cortex than in the fetal cerebellum (Figure 1E,F, Table 2). A summary of histological changes in the cerebellar cortex during embryogenesis is depicted in Figure 1G and Table 2.

### 3.2. Pleiotrophin Cell-Type Expression Changes during Cerebellar Cortex Development in Humans

The anti-PTN antibody recognizes a 168 a.a. protein, the expression and localization of which were variable through the different stages of the human fetal cerebellar cortex development evaluated (Table 3). As shown in Figure 2A,E and Table 3, PTN protein expression was lower at 17 gw, where it was immunoreactive in the nuclei and somas of cells that were vertically extended throughout the EGL. These patterns were compared to GFAP-positive stains in the EGL. At 31 gw, PTN expression was increased; it was located with high intensity in all nuclei of the EGL where GFAP staining was reduced to some glial projections (Figure 2F,J). PTN expression depicted a new decrease at 36.6 gw in the EGL, but GFAP staining showed an increase in intensity and expression (Figure 2O); PTN localization was limited to a few nuclei of the EGL and in some of the cytoplasm and nuclei of PC neurons (Figure 2K). At 37.4 gw, the intensity of PTN decreased and was only expressed in the cytoplasm of PCs, while GFAP stains showed diffuse distribution across all cerebellar cortex layers (Figure 3A). Thus, PTN intensity became barely detectable in the cytoplasm of PCs at 38 gw (Figure 3F). Interestingly, PTN expression reappeared in the cell cytoplasm of the EGL in the neonatal sample, but it disappeared in PCs (38 gw) (Figure 3K).

### 3.3. PTN Receptor Expression Spatiotemporally Changes in Cerebellar Cortex Development

As PTN receptors, the expression of PTPRZ1, ALK, and NRP-1 proteins was evaluated using immunohistochemistry. All characterizations are shown in Figure 2, Figure 3 and Figure 4.

#### 3.3.1. Gestational Week 17

At this stage, the EGL was clearly detected in cerebellar cortex tissue (Figure 1A). The immunoreactivity to PTPRZ1 was low in some cell nuclei of this layer. The PTPRZ1 protein also showed diffuse distribution and high intensity in IGL cell soma (Figure 2B, Table 4). In contrast, ALK protein staining was negative in all cerebellar cortex layers (Figure 2C, Table 5). NRP-1 expression was diffusely distributed in the neuropil zone where it had moderate intensity. The EGL was completely negative to NRP-1 expression (Figure 2D, Table 6).

#### 3.3.2. Gestational Week 31

At this stage, cell density began to increase in the EGL, and PC bodies were easily seen (Figure 1B). The intensity of PTPRZ1 expression was higher in all soma of external granular cells. PTPRZ1 was also distributed abundantly in the cytoplasm of PCs. In the IGL, PTPRZ1 was positive in the soma and nuclei of ganglion neurons (Figure 2G, Table 4). PTPRZ1-positive patterns were similar to GFAP-positive cells, showing the morphology and location of Bergmann glia (Figure 2J). In addition, ALK protein expression was only observed in the cell nuclei of the ML at this age, with high-intensity staining in most cell nuclei (Figure 2H, Table 5). The NRP-1 protein was localized in the soma of the external granule cells, too. NRP-1 expression was also shown in the neuropil zone. Its staining intensity was higher in the EGL than in the neuropil zone (Figure 2I, Table 6).

#### 3.3.3. Gestational Week 36.6

At this stage, PC body size was increased (Figure 1C). The expression of PTPRZ1 was diffusely distributed through all cerebellar cortex layers, as was a GFAP staining pattern (Figure 2L,O). In granule cells and Bergmann glia, PTPRZ1 showed nuclear and soma staining with high intensity. The PCL was positive for PTPRZ1 in the cell cytoplasm, where it retained moderate intensity. PTPRZ1 was also located in the neuropil zone (Figure 2L, Table 4). On the other hand, the ALK protein was again negative in all cerebellar cortex layers (Figure 2M, Table 5). NRP-1 expression was reduced in this gestational age, it was only positive in the cytoplasm of PC, and it showed low-intensity staining (Figure 2N, Table 6).

#### 3.3.4. Gestational Week 37.4

At this stage, PTPRZ1 and GFAP staining patterns were consistently similar (Figure 3B,E). PTPRZ1 expression was seen in the soma of external granule cells and glial projections. In the PCL, PTPRZ1 was positive in the cytoplasm of PCs, which began to demonstrate a long axon. Additionally, PTPRZ1 was diffusely distributed in the ML and neuropil zone of all cerebellar cortex layers (Figure 3B, Table 4). The ALK protein was not detected in the cerebellar cortex layers (Figure 3C, Table 5), but NRP-1 expression was increased in the cytoplasm of PCs compared to the cerebellar cortex at 36.6 wg (Figure 3D, Table 6).

#### 3.3.5. Gestational Week 38

At this stage, PTPRZ1 expression showed high-intensity staining along the axon projection of PCs, which was extended to the EGL. In the ML, PTPRZ1 continued showing diffuse distribution, as the GFAP pattern did, and it was positive at the neuropil zone of all cerebellar cortex layers. In the IGL, PTPRZ1 was observed in the soma of PCs (Figure 3G, Table 4). The ALK protein was negative in the cerebellar cortex (Figure 3H, Table 5), and NRP-1 showed enhanced staining intensity in the cytoplasm of PCs (Figure 3I, Table 6).

#### 3.3.6. Neonatal Sample

At this stage, the thickness and cell density of the EGL were gradually reduced (Table 2). The staining intensity of PTPRZ1 in the EGL was lower in neonatal tissue than in fetal tissue at 38 gw (Table 4). PTPRZ1 expression was maintained along the axon projection of PCs. It showed diffuse distribution in the ML with moderate intensity, where cell nuclei were negative, and had the same staining pattern as the GFAP protein. PTPRZ1 was also positive in the neuropil zone of all cerebellar cortices (Figure 3L,O). ALK expression was not seen in the cerebellar cortex (Figure 3M, Table 5). NRP-1 expression showed lower-intensity staining in neonatal tissue than in fetal tissue at 38 gw and was limited to the cytoplasm of PCs (Figure 3N, Table 6).

## 4. Discussion

This study described, for the first time, the expression pattern of PTN and its receptors—PTPRZ1, NRP-1, and ALK—throughout the development of the human cerebellar cortex, from 17 gw to the early neonatal stage, using an immunohistochemical technique. The results indicate that, in the cases analyzed, the localization and intensity of PTN, PTPRZ1, and NRP-1 staining changed according to the gestational weeks, but the ALK protein was only expressed in the cerebellar cortex at 31 gw. We also contrasted the cell type that expresses them using the GFAP marker as an astroglial phenotype in human fetal stages.

This work shows that PTN is predominantly expressed in the EGL at 17 gw and increases at 31 gw. This immunoreactivity retains a pattern similar to Bergmann glia only at 17 gw. This shows that, in humans, PTN could be produced by Bergmann glia at 17 gw, but granular cell precursors may be the major sources of PTN at 31 gw. Even though PTN protein expression has been described in Bergmann glia, it was only reported for the postnatal stage in rats [16]. However, a similar pattern between PTN and radial glia has been reported in the cerebral cortex of rat embryos (E13), where it seems to be related to the perpendicularly running processes of neurons [12]. The PTPRZ1 protein was observed in some granule cell precursors at 17 gw, but it has a similar expression pattern to GFAP in the ML at 31 gw. This suggests that there is continuous communication between granule cell precursors and Bergmann glial cells that could implicate the PTN–PTPRZ1 pathway, which increases at 31 gw. Additionally, Bergmann glial cells may be stimulated by other molecules to lower PTN expression in humans between 17 and 31 gw. This may be related to the expression of the PTPRZ1 protein by Bergmann glia in the latest stage of rat and mouse development [29,39].

Additionally, the ALK protein was absent at 17 gw, where PTN and PTPRZ1 were weakly expressed, too, but they were highly expressed in the ML at 31 gw. This could suggest another cell type in which ALK and PTPRZ1 are co-expressed in the ML at 31 gw. It has been demonstrated that the PTN–PTPRZ1 complex promotes ALK activation in humans [40] and ALK can stimulate its expression by promoting MYCN transcription [41]; therefore, PTN–PTPRZ1 signaling can promote ALK expression in the ML. An important role of ALK in synaptic plasticity has been described in mouse *Alk*^KO^ [42], but the activation of interconnecting pathways (RAS/ERK; JAK/STAT; and PI3K/AKT) is also known, which may activate cell proliferation [43,44,45]. The latter finding is consistent with the increase in ML thickness observed in the present study after 31 gw (peak ALK expression). In parallel, NRP-1 was only localized in the neuropil zone at 17 gw, but it was highly expressed in the soma of granule cell precursors at 31 gw. These results could represent a co-localization of PTN and NRP-1 in the EGL. Although NRP-1 is a receptor that should be observed in membrane cells, it was also described that the PTN–NRP-1 interaction could stimulate NRP-1 internalization into the cell cytoplasm [27]. These reports suggest autocrine signaling in granular cell precursors. Remarkably, the increase in protein expression at 31 gw is consistent with the peak of proliferation in the human fetal cerebellar cortex [46]. Thus, autocrine signaling activated by PTN/NRP-1 axe may be involved in granule cell growth and proliferation in the cerebellar cortex.

From 36.6 gw to the postnatal stage, the ALK protein was negative in all cerebellar cortex layers. Consistent with this result, ALK protein expression has been described in the central and peripheral nervous systems in mouse embryos [33,47], but negative expression has been shown in the cerebellum of 14-day embryos to adult mouse stages [32]. Therefore, it is noteworthy to describe the negative expression of ALK in the human cerebellum, too, as a sign of other similar embryonic processes between both species. In the same period, we described PTN and PTPRZ1 expression in the human cerebellar cortex where they were co-localized in PCs. This result could involve autocrine signaling (PTN–PTPRZ1) in PCs that do not produce ALK activation. Some mutations in ALK cause its aberrant activation, which is related to the development of neuroblastoma, a tumor originating from multipotent cells of the neural crest [48,49,50]. It is hypothesized that ALK overexpression generates a defect in the differentiation of neural crest cells in neuroblastoma [51]. Therefore, in the development of the central nervous system, the inactivation of ALK is important for the correct process during embryogenesis. Interestingly, PTPRZ1 had the same staining pattern as GFAP, which suggests that Bergmann glia express PTPRZ1, too, and have intense communication with PCs. Disturbance to PTN–PTPRZ1 signaling causes some effects in the glutamate/aspartate transporter (GLAST) on Bergmann glia and, at the same time, induces PC abnormalities in organotypic slice culture [52]. These findings support possible human intercellular communication between PC and Bergmann glia that involves PTN–PTPRZ1 signaling and regulates the morphogenesis of PC dendrites.

It is also important to consider the change in the localization of PTN expression in the postnatal stage [14]. We found negative expression of PTN in PCs but moderate intensity in precursor granular cells in the EGL. This finding was also noted by Matsumoto and collaborators in 1994, who described the extracellular expression of PTN in the EGL of the postnatal mouse cerebellum [12]. Based on Matsumoto’s results and the peak migration process required to mature the cerebellum in human and mouse development [13,53,54], PTN expression could be implicated in granular cell migration from the EGL to the IGL, which may also be related to PTPRZ1 expression in Bergmann glia. Moreover, NRP-1 expression decreased at 36.6 gw and was only localized in PCs. Even though NRP-1 has been described in the peripheral nervous system [55,56], its expression in the human cerebellar cortex has not been demonstrated before. The localization of NRP-1 described in this study suggests its implication in the PC maturation process, too. NRP-1 has been described as a neuronal axon guide, which is important for synapsis connections [57]; therefore, this function can explain why we found a gradual increase in NRP-1 intensity in PCs from 37.4 gw to the neonatal stage. During these gestational weeks, PCs need dendritic arborization and axon projection to the deep cerebellar nuclei in the adult brain [58,59]. This finding implies a more complex process in PCs during cerebellar cortex development, which could implicate the auto-stimulation of PTN in complexes with PTPRZ1 and NRP-1 receptors. The knowledge of anatomical and physiological changes in the cerebellum without these proteins is limited, but an assay using PRPTZ1 −/− mice showed a decrease in normal motor activity and balance function in comparison with mice^WT^ [60].

We are aware that this is an exploratory study based on a small number of cases, and we hypothesize rather than provide conclusive results about the role of each protein studied in human cerebellar maturation. Nevertheless, we believe that the information presented here is valuable because it provides insights into the biology of cerebellar development. In addition, since it is known that there are neoplasms originating from immature cerebella, such as medulloblastomas, these results raise the question of whether these molecules—the involvement of which has been demonstrated in other neoplasms of the nervous system—may also play a role in medulloblastomas. Further studies with larger samples are needed to deepen the investigation of signaling pathways that confirm the degree to which PTN and its receptors are involved in human cerebellar development, both prenatally and postnatally.

## 5. Conclusions

The present study represents the first spatial–temporal characterization of PTN and the expression of its receptors in the human fetal and neonatal cerebellum. PTN, PTPRZ1, and NRP-1 change their expression and localization, indicating that PTN could activate different signaling pathways and may be involved in generating intercellular communication in the fetal cerebellum and neonatal cells during their development. Thus, PTN could be involved in the morphogenesis of PCs as well as granular cell proliferation and migration in the human cerebellar cortex. Furthermore, PTPRZ1 expression in all cerebellar cortex layers suggests that these proteoglycans play an important role in the maturation of the human cerebellum. Given the exploratory nature of this work, the extent to which these molecules are expressed in immature human cerebellar tissue needs to be confirmed by using other methodological approaches and larger samples.

## Figures and Tables

**Figure 1 cells-12-01733-f001:**
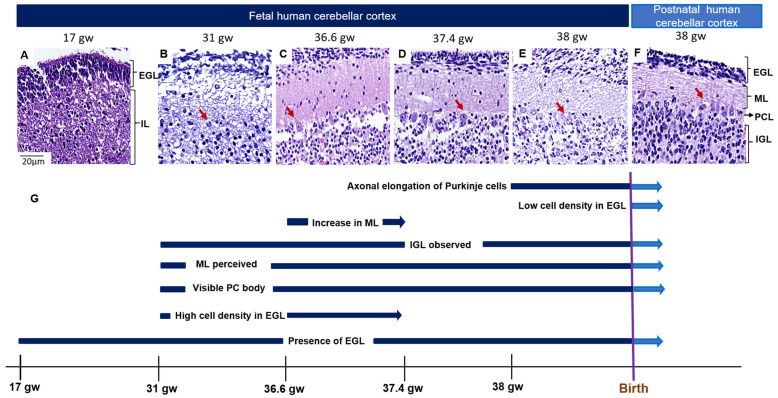
Histological changes in cerebellar cortex in human development. (**A**–**F**) Hematoxylin and eosin staining of fetal cerebellar and neonatal samples; (**G**) timeline of morphological changes in the cerebellar cortex. Dark blue—fetal samples. Light blue—neonatal sample. EGL—external granular layer; IGL—internal granular layer; IL—internal layer; ML—molecular layer; PCL—Purkinje cell layer. The scale bar in image A is 20 µm and applies to all microphotographs. The red arrows show PCs.

**Figure 2 cells-12-01733-f002:**
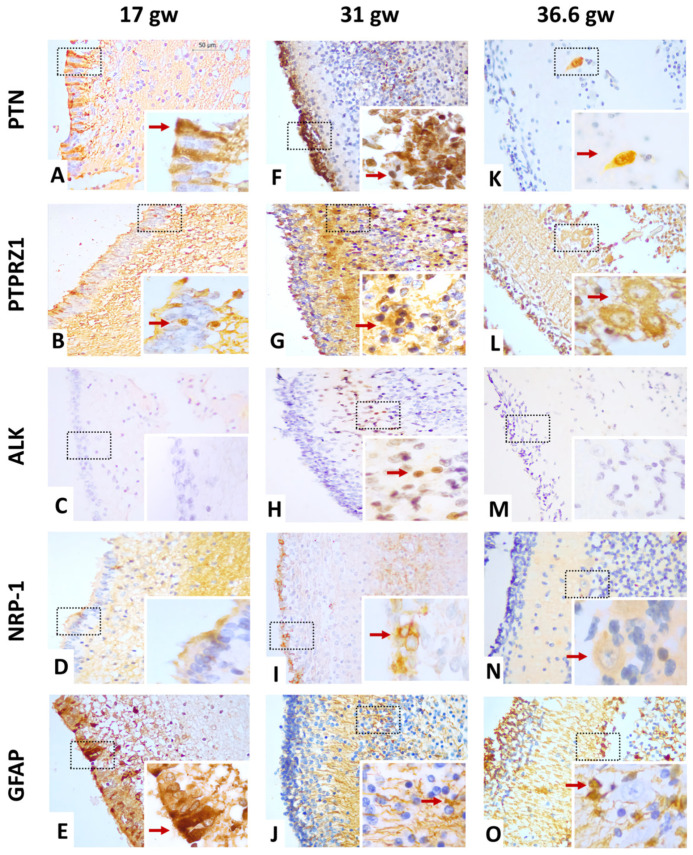
Representative image of PTN and the expression of its receptors in the human cerebellar cortex, from 17 gw to 36.6 gw. (**A**–**D**) The expression of PTN, PTPRZ1, and NRP-1 are low; ALK is negative at 17 gw. (**E**) GFAP expression is similar to PTN staining; (**F**–**I**) the expression of PTN, PTPRZ1, ALK, and NRP-1 is high at 31 gw; (**J**,**O**) GFAP pattern expression is similar to PTPRZ1 expression; (**K**–**N**) the expression of PTN and NRP-1 is limited to PCs, PTPRZ1 has diffuse distribution, and ALK is negative. The scale bar in image (**A**) is 50 µm and applies to all microphotographs.

**Figure 3 cells-12-01733-f003:**
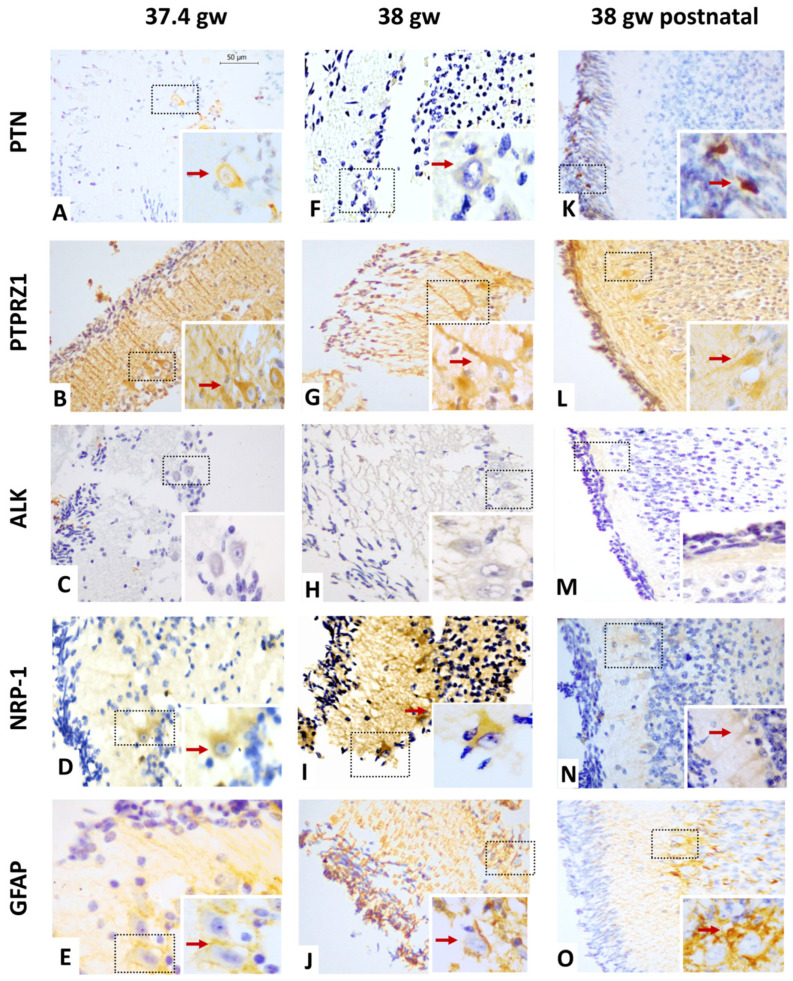
Representative image of PTN and the expression of its receptors in human cerebellar cortex from 37.4 gw to neonatal stage. (**A**–**D**) The expression of PTN and NRP-1 is limited to PCs, PTPRZ1 has a high intensity, ALK is negative at 37.4 gw; (**E**,**J**,**O**) GFAP pattern expression is similar to PTPRZ1 distribution; (**F**–**I**) PTN has a low intensity in PCs, NRP-1 has a moderate intensity in PCs, PTPRZ1 shows diffuse distribution, ALK is negative at 38 gw; (**K**–**N**) PTN expression is located in EGL, NRP-1 has a low intensity in PCs, PTPRZ1 has a high intensity in all cerebellar cortex layers; ALK is negative at the neonatal stage. The scale bar in image (**A**) is 50 µm and applies to all microphotographs.

**Figure 4 cells-12-01733-f004:**
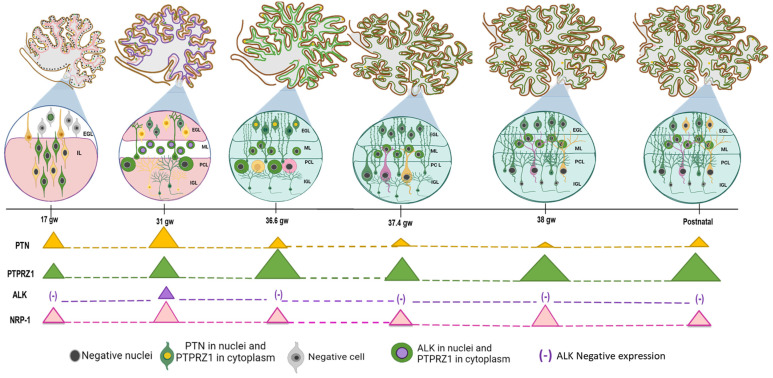
Schematic illustration of the spatiotemporal expression of PTN and its receptors from 17 gw to the neonatal sample of the human cerebellar cortex. EGL—external granular layer; IGL—internal granular layer; IL—internal layer; ML—molecular layer; PCL—Purkinje cell layer. PTN (yellow). PTPRZ1 (green). ALK (purple). NRP-1 (pink). Negative (black). The triangle size is related to protein level expression. Created on BioRender.com.

**Table 1 cells-12-01733-t001:** Clinical characteristics of human fetal and neonatal samples.

Age	Sex	Cause of Death
Fetal samples		
17 gw	Male	Interruption of placental blood flow
31 gw	Female	Congenital heart disease
36.6 gw	Male	Umbilical venous thrombosis
37.4 gw	Female	Amniotic fluid aspiration
38 gw	Female	Placental hypoxia–ischemia
Neonatal sample		
38 gw	Female	Respiratory failure

**Table 2 cells-12-01733-t002:** Morphometric parameters in cerebellar cortex development.

Gestational Weeks	EGL Thickness (µm) (Mean ± SD)	EGL Cell Density (Number/5000 µm^2^) (Mean ± SD)	PC Area (µm^2^) (Mean ± SD)	ML Thickness (µm) (Mean ± SD)
17	29.4 ± 6.3	47.0 ± 8.4	N/A	N/A
31	42.1 ± 10.9	77.3 ± 22.3	81.7 ± 25.9	48.6 ± 14.7
36.6	76.9 ± 29.3	143.0 ± 43.8	247.6 ± 44.9	111.5 ± 22.0
37.4	32.2 ± 6.3	66.3 ± 11.9	261.9 ± 11.5	54.4 ± 7.9
38	51.4 ± 9.1	81.0 ± 9.9	278.7 ± 23.5	56.6 ± 11.8
Postnatal	26.9 ± 4.1	51.5 ± 5.6	307.2 ± 30.5	49.3 ± 6.4

EGL—external granular layer; ML—molecular layer; PC—Purkinje cell; SD—standard deviation; N/A—not applicable.

**Table 3 cells-12-01733-t003:** PTN staining intensity in cerebellar cortex by gestational week.

	EGL			PCL			ML			IGL		
Staining Intensity	1+	2+	3+	1+	2+	3+	1+	2+	3+	1+	2+	3+
17 gw	5.5%	6.1%	1.7%	N/A	N/A	N/A	N/A	N/A	N/A	0%	0%	0%
31 gw	7.5%	9.8%	15.9%	0%	0%	0%	0%	0%	0%	21.3%	12.0%	6.9%
36.6 gw	24.0%	7.4%	1.8%	0%	0%	66.%	0%	0%	0%	0%	0%	0%
37.4 gw	0%	0%	0%	33.3%	0%	33.3%	0%	0%	0%	0%	0%	0%
38 gw	0%	0%	0%	33.3%	0%	0%	0%	0%	0%	0%	0%	0%
Postnatal	20.6%	8.9%	3.7%	0%	0%	0%	0%	0%	0%	0%	0%	0%

The difference in the sum of the percentages with respect to 100% in each region and gestational age corresponds to cells with no signal (0+). EGL—external granular layer; IGL—internal granular layer; ML—molecular layer; N/A—not applicable; PCL—Purkinje cell layer.

**Table 4 cells-12-01733-t004:** PTPRZ1 staining intensity in cerebellar cortex by gestational week.

	EGL			PCL			ML			IGL		
Staining Intensity	1+	2+	3+	1+	2+	3+	1+	2+	3+	1+	2+	3+
17 gw	4.3%	6.3%	2.5%	N/A	N/A	N/A	N/A	N/A	N/A	2.2%	9.3%	34.4%
31 gw	6.6%	12.7%	13.8%	0%	11.1%	22.2%	0.1%	8.1%	24.9%	4.2%	17.6%	15.2%
36.6 gw	1.8%	7.3%	24.0%	7.6%	20.4%	5.1%	5.5%	10.2%	8.8%	8.1%	13.4%	11.6%
37.4 gw	11.9%	10.3%	10.2%	0%	14.2%	19.0%	2.3%	11.9%	16.2%	7.3%	15.9%	9.6%
38 gw	24.4%	7.5%	1.3%	0%	0%	33.3%	4%	16%	13.3%	20.2%	9.4%	3.6%
Postnatal	21.8%	11.0%	0.4%	0%	9.0%	24.2%	30.7%	20%	2.6%	16.1%	15.1%	1.9%

The difference in the sum of the percentages with respect to 100% in each region and gestational age corresponds to cells with no signal (0+). EGL—external granular layer; IGL—internal granular layer; ML—molecular layer; N/A—not applicable; PCL—Purkinje cell layer.

**Table 5 cells-12-01733-t005:** ALK staining intensity in cerebellar cortex by gestational week.

	EGL			PCL			ML			IGL		
Staining Intensity	1+	2+	3+	1+	2+	3+	1+	2+	3+	1+	2+	3+
17 gw	0%	0%	0%	N/A	N/A	N/A	N/A	N/A	N/A	0%	0%	0%
31 gw	0%	0%	0%	0%	0%	0%	12.9%	14%	17.9%	0%	0%	0%
36.6 gw	0%	0%	0%	0%	0%	0%	0%	0%	0%	0%	0%	0%
37.4 gw	0%	0%	0%	0%	0%	0%	0%	0%	0%	0%	0%	0%
38 gw	0%	0%	0%	0%	0%	0%	0%	0%	0%	0%	0%	0%
Postnatal	0%	0%	0%	0%	0%	0%	0%	0%	0%	0%	0%	0%

The difference in the sum of the percentages with respect to 100% in each region and gestational age corresponds to cells with no signal (0+). EGL—external granular layer; IG—internal granular layer; ML—molecular layer; N/A—not applicable; PCL—Purkinje cell layer.

**Table 6 cells-12-01733-t006:** NRP1 staining intensity in cerebellar cortex by gestational week.

	EGL			PCL			ML			Neuropil		
Staining Intensity	1+	2+	3+	1+	2+	3+	1+	2+	3+	1+	2+	3+
17 gw	0%	0%	0%	N/A	N/A	N/A	N/A	N/A	N/A	8.2%	20.1%	4.9%
31 gw	4.4%	9.7%	18.4%	0%	0%	0%	29.1%	4.1%	0%	24.7%	6.8%	1.6%
36.6 gw	0%	0%	0%	22.5%	4.7%	0%	0%	0%	0%	0%	0%	0%
37.4 gw	0%	0%	0%	0%	25%	8.3%	0%	0%	0%	0%	0%	0%
38 gw	0%	0%	0%	0%	0%	33.3%	0%	0%	0%	0%	0%	0%
Postnatal	0%	0%	0%	30%	3.3%	0%	0%	0%	0%	0%	0%	0%

The difference in the sum of the percentages with respect to 100% in each region and gestational age corresponds to cells with no signal (0+). EGL—external granular layer; IGL—internal granular layer; ML—molecular layer; N/A—not applicable; PCL—Purkinje cell layer.

## Data Availability

The data presented in this study are available on request from the corresponding author.

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
