# Peer review of "Pleiotrophin and the Expression of Its Receptors during Development of the Human Cerebellar Cortex"

_cells, 2023, doi:10.3390/cells12131733_

Round 1
Reviewer 1 Report
In the present study, the authors performed an immunohistochemical analysis of pleiotrophin (PTN) and its receptors PTPRZ1, ALK and NRP-1 in fetal and neonatal cerebellum of humans. Based on the results, the authors conclude that PTN and its receptors could be involved in fetal and neonatal cerebellar development of humans. Although the analysis in the present study is limited to immunohistochemistry, experimental results of humans are valuable. Nonetheless, I raise the following points to be revised in the present manuscript.
Major:
As PTPRZ1 signals were intense during all the period examined, readers may be concerned whether or not non-specific staining occurred. It is preferable to address this concern. For example, the authors may be able to check whether immunohistochemistry using another anti-PTPRZ1 primary antibody results in the same results.
Minor:
1. Immunohistochemistry in Materials and Methods (page 3): Although the anti-PTN, PTPRZ1 and ALK antibodies were mouse antibodies, the anti-NRP-1 antibody was a rabbit antibody. The anti-GFAP antibody might be also a rabbit antibody. However, only an anti-mouse IgG antibody is stated as the secondary antibody in this section.
2. First paragraph in Results (page 3, line 21): I cannot understand why the percentage of females is 57.14%. If four female and two male samples were used, the percentage of females is 67%.
3. Discussion (page 9, lines 17-19): Reference number [8] should be inserted for Matsumoto et al. (1994).
English should be polished throughout the manuscript.
Reviewer 2 Report
GFAP immunoreaction is not clear.
Why didn't use also vimentin as a marker of glial cells?
How do you calculate the increase in the size of Purkinje cells?
In Fig.2 B, G, L where is positivity?
A single neonatal case is too little.
Why seminoma as tissue control for PTN?
GFAP is a marker of astrocytes, where is positivity in Fig 2 (E, J, O)?
In the boxes, the highlighted cells seem to be neurons and no glial cells.
The quality of the photos is very poor and positive cells are not present and
in Fig.3 the same.
The conclusion is to be reviewed in the context of improvement in data and several cases.
The bibliography must be integrated.
Round 2
Reviewer 1 Report
The authors have fully responded to my comments. I have no further comments.
English should be polished throughout the manuscript.
Author Response
"Please see the attachment."

Reviewer 2 Report
The bibliography needs to be improved and the quality of histological images too. It is still difficult to recognize the positivity of immunohistochemical reactions.
Author Response
"Please see the attachment."
